# Variational Diffusion Recovery with Multiscale Energy Models for Few-Step Generation

## Abstract

Energy-Based Models (EBMs) offer a principled framework for modeling complex data distributions. However, their training via contrastive divergence is often hindered by slow and unstable MCMC sampling—especially in high-dimensional settings. Recent advances such as diffusion recovery likelihood and cooperative diffusion recovery likelihood (CDRL) improve tractability by training conditional EBMs across multiple noise scales. These methods rely on Langevin sampling at each scale, translating to slow inference. In this work, we propose VDRME (Variational Diffusion Recovery with Multiscale Energy), a novel framework that amortizes MCMC sampling at each scale using conditional generators. We train conditional EBMs using a variational lower bound on the maximum likelihood objective, enabling efficient one-step sampling per scale. To further enhance diversity and prevent mode collapse, we introduce entropy-based regularization of the generators. Unlike diffusion GANs, which rely on adversarial losses and classifier guidance, VDRME maintains a fully energy-based formulation and produces informative energy priors for downstream tasks. Our experiments demonstrate that VDRME achieves fast, few-step generation with high perceptual quality, improved convergence, and strong performance on downstream tasks such as out-of-distribution detection and density estimation. By replacing score-based training with multiscale generators and avoiding traditional MCMC, VDRME offers a scalable, interpretable, and efficient alternative to existing EBMs and diffusion-based generative models.

## 1 Introduction

Energy-based models (EBMs) have long served as a powerful framework in machine learning (Song & Kingma, 2021; Yoon et al., 2023; Pang et al., 2020; Nijkamp et al., 2019), with applications spanning image generation (Gao et al., 2020b; Zhu et al., 2023), natural language processing (Gladstone et al., 2025), discriminative learning (Guo et al., 2023a; Grathwohl et al., 2019a), and reinforcement learning (Haarnoja et al., 2017). In imaging, EBMs are particularly compelling as expressive priors for inverse problems (Chand & Jacob, 2024b;a), out-of-distribution (OOD) detection (Liu et al., 2020), and compositional generation Thornton et al. (2025). Traditionally, EBMs are trained using contrastive divergence (CD) (Du et al., 2021; Wang et al., 2022), which maximizes the energy gap between real and synthesized samples (Gao et al., 2020b; Zhu et al., 2023; Grathwohl et al., 2019a; Song & Kingma, 2021). Despite its theoretical appeal, CD training is computationally intensive due to the reliance on iterative Markov chain Monte Carlo (MCMC) sampling to synthesize fake samples. Moreover, the adversarial training strategy often suffers from instability.

Diffusion models offer an alternative by leveraging score-based training (Ho et al., 2020a; Song et al., 2020a;b; Kim et al., 2021), where the conditional distributions are learned across noise levels in a diffusion process. These models only learn the gradient of the log condtional densities (score), which are then used to synthesize data via ancestral sampling from Gaussian noise. Diffusion models can be interpreted as a time-indexed sequence of energy-based models, where the score function corresponds to the negative gradient of an underlying energy function. Diffusion models do not explicitly constrain the score to be the gradient of an energy function; parameterizing the score as the gradient of an EBM offers a compelling alternative, and satisfies several desirable properties

including conservative nature of score and ability to compose multiple EBMs to obtain a composite EBM at inference time Thornton et al. (2025). In addition, diffusion models require an expensive ODE to estimate the likelihood, unlike EBMs that explicity provide the likelihood of a sample; this property makes EBM attractive in downstream tasks including out of domain (OOD) estimation.

Several efforts have been directed to improve the sample quality of EBMs in the recent years. For instance, Diffusion Recovery Likelihood (DRL) (Gao et al., 2020b) proposed to train a sequence of EBMs for the marginal distributions of the diffusion process, motivated by diffusion models. DRL simplifies the EBM training by focusing on conditional distributions (conditioned on the previous noise scale), which are easier to sample from than marginals. DRL relaxes the assumption of the reverse diffusion process being a Gaussian that is commonly made by diffusion models, which are only valid when the step-size is small. This relaxation allows the use of significantly fewer noise scales (6-8) than conventional diffusion models. Recently, EBMs that can generate high-quality samples that closely match state-of-the-art diffusion models were introduced in Thornton et al. (2025); they relied on initialization of the EBM model using pretrained diffusion models, and a distillation-based training strategy.

Recent years have seen rapid advancements in reducing the sampling burden in diffusion and flow models. The approaches including distillation (Kim et al., 2025), consistency models (Song et al., 2023), and mean flow models (Geng et al., 2025) learn deterministic paths from an initial prior density to the data distribution, enabling impressive single or few-step sampling. In a parallel effort, several authors have proposed to combine diffusion models with adversarial training strategies (e.g. UFOGen (Xu et al., 2024)) and initialization using pretrained diffusion models to offer few or even single-step image generation. Despite the advances in EBM models described above, the area of few-step generation remains largely unexplored with EBMs. For instance the DRL approach still needs about 30 Langevin steps for each of the 6-8 noise scales. While cooperative Diffusion Recovery Likelihood (CDRL) (Zhu et al., 2023) enhances efficiency by introducing amortized initializers, they still require 15 Langevin steps per scale. While the quality of samples generated has improved significantly with distillation, the number of function evaluations (NFE) required is still substantially higher than in the most recent diffusion and flow-based methods Thornton et al. (2025). The main focus of this work is to introduce a principled EBM model for few-step image generation, offering a scalable and efficient alternative that unifies EBMs with modern diffusion frameworks.

We propose **VDRME**, a novel framework bridging EBMs and diffusion models to reduce sampling burden. Our approach is based on the reinterpretation of the maximum likelihood estimation as a bi-level variational optimization problem (Grathwohl et al., 2020). This formulation amortizes costly MCMC sampling at each scale using a generator, which takes samples from the previous scale and noise to create current scale samples. The above works Grathwohl et al. (2020) utilized this approach in a single-scale generation setting, akin to a traditional GAN. In contrast, our method tackles a multiscale setting; generating samples at each scale from previous scale ancestors is simpler than from pure noise; this sequential process makes training more stable and less prone to mode collapse. With the assistance of the conditional energy with diffusion recovery likelihood and entropy estimator, the training and sampling process are more tractable. Unlike diffusion GANs, which rely on adversarial losses and classifier guidance, VDRME maintains a fully energy-based formulation and produces informative energy priors for downstream tasks.

Our contributions are as follows:

- We reformulate the maximum likelihood training of the conditional energy models at each scale of the DRL model as a bi-level variational optimization (Grathwohl et al., 2020), involving a conditional generator. This replacement enables us to significantly reduce the sampling burden of DRL.

- We regularize the generator at each scale using an entropy regularization term, which stabilizes the optimization and enhances sample diversity.

- We show that VDRME achieves state-of-the-art performance among EBMs on benchmarks such as CIFAR-10 and ImageNet 256, while maintaining lower sampling computational cost than current EBM methods.

- We demonstrate the versatility of our framework in density estimation and out-of-distribution detection.

## 2 BACKGROUND

### 2.1 DIFFUSION AND ENERGY-BASED MODELS

Diffusion models constitute a prominent family of generative models, where noise is gradually injected through a forward diffusion process over T steps with variance schedule $\{\beta_t\}$. Specifically, the transition diffusion forward distribution is defined as

$$x_{i+1} \sim p(x_{i+1}|x_i) := \mathcal{N}(x_i; \sqrt{\alpha_{t_i}}x_i, \beta_{t_i} I); \quad i = \{0, .., K-1\} \tag{1}$$

where $\alpha_t = 1 - \beta_t$. The noisy image $x_i$ can be also be expressed in closed form in terms of the clean image $x_0$: $x_i \sim q(x_i|x_0) := \mathcal{N}(x_i; \sqrt{\tilde{\alpha}_t}x_0, \tilde{\beta}_t I)$, where $\tilde{\alpha}_t = \Pi_{i=0}^{K-1}(1-\beta_i)$ and $\tilde{\beta}_t = 1 - \tilde{\alpha}_t$.

Diffusion models aim to learn the reverse process, or equivalently the conditional distributions $p_\theta(x_i|x_{i+1})$. Score-based diffusion models train a neural network $\epsilon_\theta(x_t, t)$ to predict the noise $\epsilon$ added during the forward process:

$$\mathcal{L}_{\text{simple}} = \mathbb{E}_{x_0,\epsilon,t}\left[\left\|\epsilon - \epsilon_\theta\left(\sqrt{\alpha_t}x_0 + \sqrt{1-\alpha_t}\epsilon, t\right)\right\|^2\right]. \tag{2}$$

The above approach makes the assumption that the backward diffusion process, or equivalently the conditional distribution $p_\theta(x_i|x_{i+1}) = \mathcal{N}(x_i|\overline{x_i}, \tilde{\beta}_i I)$ is Gaussian; this enables the approximation of the mean $\overline{x_i}$ using of Tweedies formula. Unfortunately, this approximation holds only when the step size $t_{i+1} - t_i$ is small, which translates to long MCMC chains, and hence high computational complexity, during inference.

Energy-based models (EBMs) define a probability distribution through an unnormalized energy function. For a random variable $x \sim p_{\text{data}}$, an EBM model can be explicitly denoted as:

$$p_\theta = \frac{e^{-E_\theta(x)}}{Z_\theta}, \tag{3}$$

where $E_\theta$ is a parameterized nonlinear function (often implemented as a neural network), and the normalization constant $Z_\theta = \int_X E_\theta(x)dx$ ensures a well-defined probabilistic model. The classical approach is the maximum likelihood training $\theta = \arg\max_\theta \log p_\theta(x)$, which translates to the minimization of the loss :

$$\theta = \arg\min_\theta \ \left(\mathbb{E}_{x\sim p_{\text{data}}}\left[E_\theta(x)\right] - \mathbb{E}_{x\sim p_\theta}\left[E_\theta(x)\right]\right) \tag{4}$$

The second term involves samples from the model distribution $p_\theta$, which are derived using Langevin iterations; the long MCMC sampling chain that is needed often contributes to high computational complexity of the algorithm. While several approximations (e.g. replay buffer as in Du et al. (2021)) may be used, they may also contribute to training instability in the above adversarial training strategy.

### 2.2 DIFFUSION RECOVERY LIKELIHOOD

Unlike conventional EBMs that directly model the data distribution as in equation 3, diffusion recovery instead learns the conditional distribution of the form $p_\theta(x_i|x_{i+1})$ in terms of $p(x_i)$. With the forward diffusion in equation 1, the conditional density can be obtained using Bayes' rule Gao et al. (2020b) as

$$p_\theta(x_i|x_{i+1}) = \frac{1}{Z_\theta(t_i)} \exp(-E_\theta(x_i, t_i)) \ \exp\left(-\frac{\|\sqrt{\alpha_{t_{i+1}}} \, x_{t_i} - x_{i+1}\|^2}{2\beta_{t_i}}\right), \tag{5}$$

where $E_\theta(x_i, t_i)$ is the energy corresponding to the marginal distribution $p_\theta(x_i, t_i)$. The multiplication by the second Gaussian term attenuates regions of $p_t$ away from the mean $x_{i+1}/\sqrt{\alpha_t}$. The conditional EBM at each time scale is given by

$$\mathcal{E}_\theta(x_i|x_{i+1}, t_i) = E_\theta(x_i, t_i) + \frac{\|\sqrt{\alpha_{t+1}} \, x_{t_i} - x_{i+1}\|^2}{2\beta_{t_i}} \tag{6}$$

DRL relies on ancestral sampling, proceeding through a sequence of samples $x_{t_i} \sim p_\theta(x_i); i = K-1, .., 0$. At each scale, it relies on Langevin sampling to derive samples from $p_\theta(x_i)$. The total

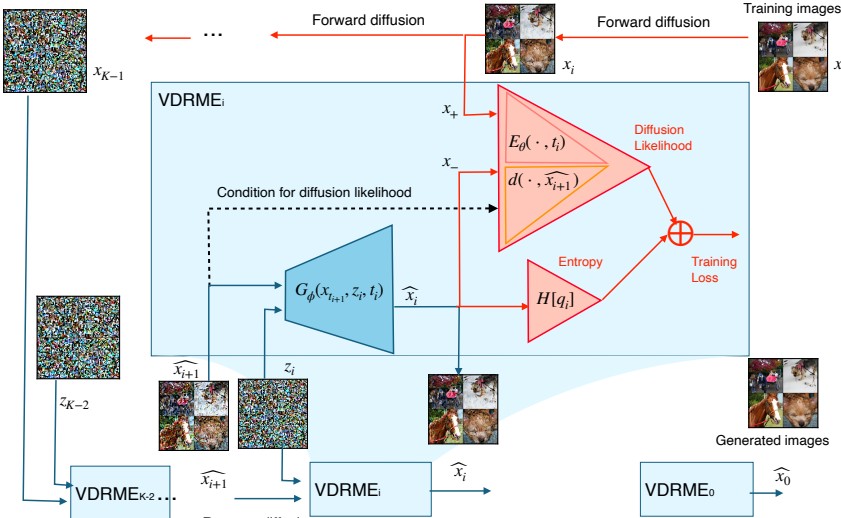

Figure 1: Overview of the VRDME training and inference: The goal is to train a few-scale energy-based model $E_\theta(.,t_i)$. Current multi-scale models use ancestral sampling using Langevin MCMC at each scale, resulting in slow inference. We amortize the Langenvin MCMC using conditional generators, thus reducing the NFE to the number of scales. The $i^{\text{th}}$ generator receives images from the $(i+1)^{\text{th}}$ generator, together with a noise input $z_i$. The generator outputs can yield samples non-Gaussian conditional distributions. Score-based diffusion models assume the reverse diffusion to be Gaussian, which is only valid when the step size is small; our formulation thus enables the use of larger steps, thus reducing the number of steps. The components used only during training are shown in red. During training, the output of the $i^{\text{th}}$ generator denoted by $\widehat{x}_i$ is fed to the diffusion likelihood loss equation 6, which uses the generated images from the previous scale $\widehat{x}_{i+1}$ as condition. The total loss at each scale is the difference of the diffusion likelihood between the true images $x_i$ and the generated images $\widehat{x}_i$ and the entropy estimate equation 14, as shown in equation 10. Entropy regularization minimizes the risk of mode collapse and training instabilities.

number of sampling steps are thus determined by the number of scales $K$, as well as the number of Langevin steps at each iteration. DRL uses around 6 scales and 30 MCMC steps per scale Gao et al. (2020c). An initializer was used in CDRL to reduce to 15 MCMC steps per scale. The main focus of this work is to amortize the expensive MCMC steps at each scale using a entropy regularized network, thus significantly reducing the computational complexity of training and inference.

## 3 VARIATIONAL DIFFUSION RECOVERY WITH MULTISCALE ENERGY

### 3.1 VARIATIONAL APPROXIMATION OF MCMC SAMPLING

By extending equation 4, the maximum likelihood training of the sequence of EBMs can be formulated as the minimization of the loss function

$$L(\theta) = \sum_{i=0}^{K-2} \left( \mathbb{E}_{x_i \sim p(x_i|x_{i+1})} \left[ \mathcal{E}_\theta(x_i|x_{i+1}) \right] + \log Z_\theta(t_i) \right) \tag{7}$$

DRL uses equation 20 for the second term to obtain

$$\theta^* = \arg\min_\theta \sum_{i=0}^{K-2} \left( \mathbb{E}_{x_i \sim p_{x_i|x_{i+1}}} \left[ E_\theta(x) \right] - \mathbb{E}_{x \sim p_\theta} \left[ E_\theta(x) \right] \right) \tag{8}$$

which involves costly Langevin dynamics. Instead, we utilize a sequence of generators to sample from the conditional densities at each energy scale, replacing iterative MCMC updates with amortized inference.

We focus on the second term in equation 7 for a specific $t_i$. The result Grathwohl et al. (2020) demonstrates that the second term has a lower bound based on an approximation $q_\phi(x_i|x_{i+1},t_i)$ of

the true conditional distribution $p(x_i|x_{i+1})$:

$$\log Z_\theta(t_i) = \max_\phi \big( - \mathbb{E}_{x_i \sim q_\phi(x_{i+1}, t_i)}[\mathcal{E}_\theta(x_i|x_{i+1}, t_i)] + H[q_\phi(x_{i+1}, t_i)] \big). \tag{9}$$

See Appendix for details. The last term is the entropy of $q_\phi$. The equality holds when $p_\theta = q_\phi$. Combining with equation 7, the joint optimization scheme is given by

$$\{\theta^*, \phi^*\} = \arg\min_\theta \max_\phi \sum_{i=0}^{K-2} \Big( \mathbb{E}_{p(x_i|x_{i+1})}[\mathcal{E}_\theta(x_i|x_{i+1}, t_i)] - \mathbb{E}_{q_\phi(x_{i+1}, t_i)}[\mathcal{E}_\theta(x_i|x_{i+1}, t_i)] + H_q(x_{i+1}, t_i) \Big) \tag{10}$$

The above result can be seen as a generalization of the single-scale approach in Grathwohl et al. (2020) to the multiscale setting. We model the samples $\widehat{x}_i$ from $q_\phi(x_i|x_{i+1}, t_i)$ as

$$\widehat{x}_i = G(x_{i+1}, z_i, t_i); \;\; z_i \sim \mathcal{N}(0, I); \; i \in [0, K-2] \tag{11}$$

We denote these generated or fake samples by $\widehat{x}_i$ to differentiate them from the true samples $x_i$ generated by equation 1. Here, $G$ is a scale-dependent generator that takes in $x_{i+1}$ and a Gaussian random variable $z_i$, approximating the MCMC sampling process from the distribution $p(x_i|x_{i+1})$. The entropy term in the generator promotes diverse samples and utilizes the noise input. The overall optimization is framed as a joint min-max game, akin to GAN models, encompassing all diffusion steps. This adversarial variational objective enables training EBMs without explicit Langevin sampling while ensuring high-dimensional data sampling, diversity, and stability.

This formulation resembles diffusion-GAN methods such as UFOGen, but differs on multiple fundamental terms. UFOGen models $\widehat{x}_i \sim \mathcal{N}(x_i; \sqrt{\bar{\alpha}}\widehat{x}_0, \tilde{\beta}_t I)$[1], which is a Gaussian approximation of $p(x_i|x_{i+1})$ that is only valid for small steps. By constrast, our generator eliminates the Gaussian restriction, resulting in improved quality, as seen from our results. Diffusion GAN models often minimize the Jensen-Shannon divergence between $p(x_i|x_{i+1})$ and $q_\phi(x_i|x_{i+1}, t_i)$ at each scale, together with a KL divergence loss on the marginals Xu et al. (2024). In contrast, our loss at each scale resembles Wasserstein GAN (WGAN) loss. However, we do not constrain the classifier's Lipschitz constant, which typically aids the WGAN training stability. Instead, we maximize the generators' entropy as a regularizer to reduce mode collapse risk. Additionally, the quadratic likelihood term emphasizes specific regions of the probability densities, further mitigating the risk of mode collapse.

### 3.2 ENTROPY ESTIMATION

To estimate the entropy term $H_{q_\phi}$ in Equation 10, we employ two non-parametric estimators: a 1-nearest-neighbor (1-NN) estimator Lombardi & Pant (2016) for the entropy for low-dimensional data (e.g., checkerboard experiment) and a mutual-information approximation for high-dimensional cases such as images. In the 1-NN estimation, we generate two set of samples from $q_\phi(x_i|x_{i+1})$, denoted by $(\widehat{x}_i)_1$ and $(\widehat{x}_i)_2$ using the generator by fixing the condition $x_{i+1}$ and by varying the noise realizations. We use the following relation where $d$ is the dimension Lombardi & Pant (2016):

$$\hat{H}_{1NN} = d \log \Big( 2\|(\widehat{x}_i)_1 - (\widehat{x}_i)_2\| \Big) + c, \tag{12}$$

where $c$ is a constant. Intuitively, entropy maximization will encourage $(\widehat{x}_i)_1$ to be separated from $(\widehat{x}_i)_2$, thus preventing from collapsing to a single point.

In higher dimensions, we maximize the entropy of $q_\phi$ using the method in Kumar et al. (2019). Assuming $x_{i+1}$ is deterministic in equation 11, the entropy of $\widehat{x}_i \sim q_\phi(x_{i+1}, t_i)$ is defined by

$$
\begin{aligned}
H(\widehat{x}_i|x_{i+1}) &= I(\widehat{x}_i, z_i|x_{i+1}) + H(\widehat{x}_i|z_i, x_{i+1}) = I(\widehat{x}_i, z_i|x_{i+1}) + \underbrace{H\Big( G(x_{i+1}, z_i, t_i)|z_i, x_{i+1}\Big)}_{=0} \\
&= I(x_i, z_i|x_{i+1}),
\end{aligned}
\tag{13}
$$

where $I(x_i, z_i|x_{i+1})$ is the mutual information between $x_{t_i}$ and $z_i$. The last term in the first equation is zero because, given $x_{i+1}$ and $z_i$, the generation process is deterministic. Thus, entropy estimation reduces to the mutual information between generated samples $x_{t_i}$ and $z_i$; maximizing the mutual

---

[1]Here, $\widehat{x_0} = (x_{t_{i+1}} - \epsilon_\phi(x_{t_{i+1}}), t_{i+1})/\sqrt{\tilde{\alpha}_{t_i}}$

information between $x_{t_i}$ and $z_i$ in equation 11 ensures the generator accounts for its noise input $z$. We approximate mutual information using Jensen-Shannon divergence between the joint distribution $p(x_i, z_i)$ and the product of the marginals $p(x_i)p(z_i)$. Specifically, we form positive pairs $(x_i, z_i)$ and negative pairs by permuting samples in each batch. The estimator is approximated as $H(x_i|x_{i+1}) \approx I_{JSD}(x_i, z_i)$, where:

$$I_{JSD}(x_i, z_i) = \sup_\psi \Big( \mathbb{E}_{p(x_i,\, z_i|x_{i+1})}[-\mathrm{sp}(-T_\psi(x_i, z_i|x_{i+1}))] - \mathbb{E}_{p(x_t|x_{i+1})p(z_i)}[\mathrm{sp}(T_\psi(x_i, z_i|x_{i+1}))] \Big),$$

(14)

where sp is the soft-plus function and $T_\psi$ is a trainable network parameterized by $\psi$. We simulate the samples from $p(x_t|x_{i+1})p(z)$ by a random permutation of the $z$ samples in the batch.

### 3.3 GENERATOR PARAMETERIZATION

We adopt the DDPM-style parametrization with a variance scheduler $\beta_t$ mentioned above, where $\sqrt{\beta}_{t_i} = \sqrt{1 - \alpha_{t_i}}$, and $\sqrt{\alpha_{t_i}} = \sqrt{\frac{\tilde{\alpha}_{t_{i+1}}}{\tilde{\alpha}_{t_i}}} = \sqrt{\frac{1-\tilde{\beta}_{t_{i+1}}}{1-\tilde{\beta}_{t_i}}}$. We use K EBMs/generators indexed by $\{i = 0, .., K-1\}$, with boundary $t_K = T = 999$, $t_0 = 0$. In this work, we used a DDPM backbone, which uses the formulation $x_{t_i} = \sqrt{\tilde{\alpha}_{t_i}}\, x_0 + \sqrt{1 - \tilde{\alpha}_{t_i}}\epsilon$; where $\epsilon \sim \mathcal{N}(0, I)$ To benefit from the inductive bias of the DDPM backbone, we implement the generator using a noise predictor $\epsilon_\phi(x_{i+1}, z_i, t_i)$ as

$$x_{t_i} = \underbrace{\frac{1}{\sqrt{1-\beta_{t_i}}}\Big(x_{t+1} - \sqrt{\tilde{\beta}_{t_{i+1}}}\epsilon_\phi(x_{i+1}, z_i, t_i)\Big) + \sqrt{\tilde{\beta}_{t_i}}\epsilon_\phi(x_{i+1}, z_i, t_i)}_{G_\phi(x_{t_{i+1}, z_i, t_i})}$$

(15)

While the above formulation is adapted from DDPM, there are key differences. There is no additive random noise term in the above expression as in DDPM. By contrast, a random noise term $z_i$ is injected into the noise predictor $\epsilon_\phi$. This approach allows the non-linear generator to model the non-Gaussian nature of the distribution $q(x_i|x_{i+1})$. Note that DDPM models the reverse diffusion process to be Gaussian, resulting in a large number of steps.

### 3.4 IMPLEMENTATION DETAILS

Combining equation 10 with equation 14, the complete loss is specified by

$$L(\theta, \phi) = \sum_{i=0}^{K-2} \Big( \mathbb{E}_{p(x_i|x_{i+1})}[\mathcal{E}_\theta(x_i|x_{i+1}, t_i)] - \mathbb{E}_{q_\phi(x_{i+1}, t_i)}[\mathcal{E}_\theta(x_i|x_{i+1}, t_i)] + I_{\mathrm{JSD}}(x_i, z_i) \Big) \quad (16)$$

Similar to GAN training, we alternate training generators and energy models.

Generator update: Generator parameters can be optimized as

$$\phi^* = \arg\max_\phi \sum_{i=0}^{K-2} \Big( -\mathbb{E}_{q_\phi(x_{i+1}, t_i)}[\mathcal{E}_\theta(x_i|x_{i+1}, t_i)] + I_{\mathrm{JSD}}(x_i, z_i) \Big) \quad (17)$$

$$= \arg\min_\phi \sum_{i=0}^{K-2} \Big( \mathbb{E}_{z_i \sim p(z)}\big[\mathcal{E}_\theta(G(x_{t_{i+1}}, z_i, t_i)\big] - I_{\mathrm{JSD}}(G(x_{t_{i+1}}, z_i, t_i), z_i) \Big) \quad (18)$$

Thus, the generators $G(x_{t_{i+1}}, z_i, t_i)$ aim to minimize the diffusion likelihood losses specified by equation 6, while maximizing the Jensen-Shannon divergence between the joint and product distributions. The second term encourages the generators to generate diverse samples, which amounts to a smoother distribution $q_\phi(t)$ that serves as a regularization to prevent mode-collapse and improve stability.

EBM update: Because the JSD term and the likelihood terms are independent of the EBM parameters, they can be updated as

$$\theta = \arg\min \sum_{i=0}^{K-2} \Big( \mathbb{E}_{x_{t_i} \sim p(x_i|x_{i+1})}[E_\theta(x_i, t_i)] - \mathbb{E}_{z_i \sim p(z)}[E_\theta(G_\theta(x_{t_{i+1}}, z_i, t_i))] \Big) \quad (19)$$

The EBM at each scale $i$ is trained to decrease the energy of real samples while increasing the energy of generated samples. To account for $x_{t_i} \sim p(x_i|x_{i+1})$, we derive them as $x_{t_i} = \sqrt{\tilde{\alpha}_{t_i}}\, x_0 + \sqrt{1 - \tilde{\alpha}_{t_i}}\epsilon$ and $x_{i+1} = \sqrt{\tilde{\alpha}_{t_{i+1}}}\, x_0 + \sqrt{1 - \tilde{\alpha}_{t_{i+1}}}\epsilon$ with the same batch of $x_0$ and noise realizations $\epsilon$.

---

**Algorithm 1** VDRME Training

---

**Input:** Number of scales: K, variance schedules $\alpha_{t_i}, \beta_{t_i}; i = 0, .., K - 1$, Energy network $E_\theta, \epsilon_\phi$ that defines the generator, initialized by DDPM weights, entropy estimator $T_\psi$
**Hyperparameters:** Weights $\lambda_e = 1.$ and $\lambda_p = 0.01$.
**Output:** Parameters $\theta, \& \phi$.
**while** True **do**
    Sample pairs $(x_{i+1}, x_i)$
    Sample $\widehat{x}_i \sim q_\phi(x_i|x_{i+1}, t_i)$ using equation 15 and a noise realization $z_i$.
    Update $\theta$ using equation 19, assuming $\psi$ and $\theta$ to be fixed
    Permute noise realizations in the batch to evaluate JSD loss using equation 14
    Update $\phi$ using equation 18, assuming $\theta$ and $\psi$ to be fixed.
    Update $\psi$ to maximize equation 14, assuming $\theta$ and $\phi$ to be fixed
**end**

---

## 4 EXPERIMENTS

This section demonstrates the VDRME framework's rapid sampling capability and its applications. We present samples generated by the VDRME framework on CIFAR-10 and ImageNet-256 in Section 4.2 and Section 4.3, evaluated mainly using Fréchet Inception Distance (FID). We also highlight the versatility of the learned energy priors in downstream tasks such as density estimation and out-of-distribution detection (Section 4.1 and 4.4). Our results indicate that the energy model performs competitively and often surpasses other methods on both tasks. Additional samples, comparisons, and experimental details are in the Appendix.

Figure 2: Unconditional samples generated by VDRME trained on CIFAR-10 (unconditional training) using 8 sampling steps. The model achieves an FID score of 6.90.

### 4.1 VERIFICATION IN A TOY DATASET

We perform the classical checkerboard experiment to validate the approach in a low-dimensional setting. Failures in this task are often attributed to mode collapse and an irregular energy landscape. As discussed previously, we estimated the entropy term using nearest-neighbor estimation. As seen from Figure 3, the approach can provide an accurate estimate of the density and provide evenly spaced samples, even with five-step sampling.

### 4.2 UNCONDITIONAL GENERATION

We evaluate our method on CIFAR-10 (32×32) in the unconditional generation setting. The results are summarized in Table 1. Our approach achieves a few-step generation with competitive image quality, outperforming existing EBM methods without relying on teacher models or distillation losses. We note that the performance improves consistently as sampling steps increase from 4 to 8; the samples with 8 steps are shown in Figure 2, demonstrating sharp and diverse generations across categories. Importantly, our model achieves these results with a significant reduction in the number of function evaluations (NFE), which are comparable to state of the art diffusion and flow-based models, highlighting its improvement relative to prior EBMs. To stabilize training, we initialize the backbone with a pre-trained U-Net from DDPM (Ho et al., 2020a). While CDRL Zhu et al. (2023) offers better FID than our method with NFE=90, the quality of the samples deteriorate with decreas-

Table 1: Comparison of FID scores and sampling steps on CIFAR-10 generation. Our approach achieves high image quality with the fewest sampling steps among the BEMs. The models cover the training with score-matching, contrastive divergence and distillation methods.

| Models | Steps | FID ↓ |
|---|---|---|
| **Explicit EBM based method** | | |
| NT-EBM (Nijkamp et al., 2020) | - | 78.12 |
| LP-EBM (Pang et al., 2020) | 40 | 70.15 |
| Adaptive CE (Xiao & Han, 2022) | 40 | 65.01 |
| JEM (Grathwohl et al., 2019b) | - | 38.40 |
| EBM-IG (Du & Mordatch, 2019) | - | 38.20 |
| EBM-FCE (Gao et al., 2020a) | - | 37.30 |
| CoopVAEBM (Xie et al., 2021) | 15 | 36.20 |
| Divergence Triangle (Han et al., 2020) | - | 30.10 |
| VARA (Grathwohl et al., 2020) | - | 27.50 |
| EBM-CD (Du et al., 2020) | 40 | 25.10 |
| GEBM (Arbel et al., 2020) | - | 19.31 |
| HAT-EBM (Hill et al., 2022) | 50 | 19.30 |
| CF-EBM (Zhao et al., 2020b) | 60 | 16.71 |
| CoopFlow (Xie et al., 2022) | 30 | 15.80 |
| CLEL-base (Lee et al., 2023a) | 600 | 15.27 |
| VAEBM (Xiao et al., 2020) | 16 | 12.16 |
| CDRL (Guo et al., 2023b) | 18 | 9.67 |
| DRL (Gao et al., 2020c) | 180 | 9.58 |
| CLEL-large (Lee et al., 2023a) | 1200 | 8.61 |
| CDRL (Guo et al., 2023b) | 90 | 4.31 |
| Distilled-EnergyDiffusion (Thornton et al., 2025) | 35 | 3.01 |
| **VDRME (Ours)** | **4** | **7.83** |
| **VDRME (Ours)** | **6** | **7.62** |
| **VDRME (Ours)** | **8** | **6.90** |

Table 2: Comparison of FID scores and sampling steps on ImageNet-256p generation. Our method achieves the best performance among EBMs and remains competitive with GAN-, diffusion-, and autoregressive-based models. We also evaluate the effects of entropy regularization and noise input using FID scores.

| Type | Model | Steps ↓ | FID ↓ |
|---|---|---|---|
| GAN | UFOgen (Xu et al., 2024) | 4 | 7.87 |
| GAN | BigGAN (Brock et al., 2018) | 1 | 6.95 |
| GAN | GigaGAN (Kang et al., 2023) | 1 | 3.45 |
| GAN | StyleGAN-XL (Sauer et al., 2022) | 1 | 2.30 |
| DM | ADM (Dhariwal & Nichol, 2021) | 250 | 10.94 |
| DM | LDM (Rombach et al., 2022) | 250 | 3.60 |
| DM | VDM++ (Kingma & Gao, 2023) | 250 | 2.40 |
| DM | DiT (Peebles & Xie, 2023) | 250 | 2.27 |
| AR | VQVAE2 (Razavi et al., 2019) | 5120 | 31.11 |
| AR | VQGAN (Esser et al., 2021) | 256 | 18.65 |
| AR | RQTran (Lee et al., 2022) | 68 | 7.55 |
| AR | Step Distillation (Salimans & Ho, 2022) | 4 | 10.92 |
| AR | VAR (Tian et al., 2024) | 10 | 1.97 |
| AR | MAR (Li et al., 2024) | 32 | 1.93 |
| EBM | EBM-IG-128 (Du & Mordatch, 2019) | – | 43.70 |
| EBM | HAT-EBM-128 (Hill et al., 2022) | 50 | 29.37 |
| EBM | EGC (Guo et al., 2023a) | 1000 | 6.05 |
| EBM | **ADRME (Ours)** | **4** | **4.62** |

| Noise input | Entropy Loss | FID ↓ |
|---|---|---|
| ✕ | ✓ | 6.43 |
| ✓ | ✕ | 5.79 |
| ✓ | ✓ | 4.62 |

ing NFE. Similarly, the recent distilled EBM Thornton et al. (2025) also requires significantly more steps to achieve a low FID.

## 4.3 CONDITIONAL GENERATION

We evaluate conditional generation on the large-scale ImageNet benchmark using a latent EBM. We use the stable diffusion variational autoencoder (Rombach et al., 2022) to derive the latents. We adopted the pretrained DiT architecture from (Peebles & Xie, 2023) as the backbone for the denoiser. We added some initial layers to this architecture, which takes in the concatenated image $x_{i+1}$ and the noise $z_i$ and returns a single image to the backbone. The examples shown in Figure 5 correspond to class conditional generation, without any guidance term. Each row corresponds to a specific class in ImageNet. The qualitative results are reported in Table 2. To the best of our knowledge, this is the first study to evaluate few-step generation within EBMs for image sizes over 256, establishing a reference point for future EBM development.

Table 3: Comparison of AUROC scores for OOD detection based on CIFAR-10 without further fine-tuning VDRME. Our results demonstrate superior performance.

| | CIFAR-10 interp. | CIFAR-100 | CelebA | SVHN |
|---|---|---|---|---|
| PixelCNN (Salimans et al., 2017) | 0.71 | 0.63 | – | 0.32 |
| GLOW (Kingma & Dhariwal, 2018) | 0.51 | 0.55 | 0.57 | 0.24 |
| NVAE (Vahdat & Kautz, 2020) | 0.64 | 0.56 | 0.68 | 0.42 |
| EBM-IG (Du & Mordatch, 2019) | 0.70 | 0.50 | 0.70 | 0.63 |
| VAEBM (Xiao et al., 2020) | 0.70 | 0.62 | 0.77 | 0.83 |
| EBM-CD (Du et al., 2020) | - | 0.53 | 0.54 | 0.78 |
| CLEL (Lee et al., 2023b) | 0.72 | 0.72 | 0.77 | 0.98 |
| DRL (Gao et al., 2020c) | – | 0.44 | 0.64 | 0.88 |
| CDRL (Zhu et al., 2023) | 0.75 | 0.78 | 0.84 | 0.82 |
| **VDRME (Ours)-S4** | **0.72** | **0.55** | **0.86** | **0.84** |
| **VDRME (Ours)-S8** | **0.72** | **0.54** | **0.84** | **0.92** |

We perform an ablation study reported in the bottom of Table 2, where we evaluate the benefit of the different components. We report the performance of the model with no noise input, with no entropy term, and with both noise input and entropy term. The results show that both components are essential for high-quality generation.

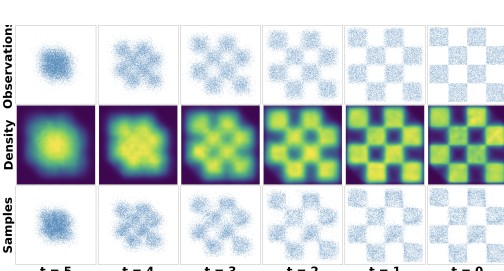
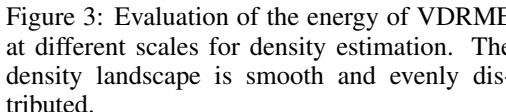
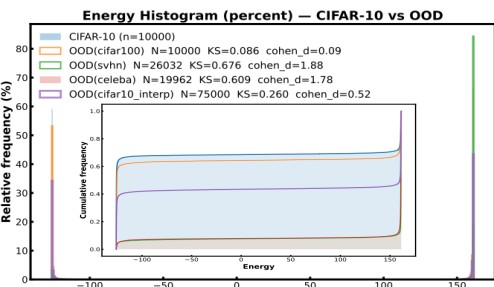

Figure 3: Evaluation of the energy of VDRME at different scales for density estimation. The density landscape is smooth and evenly distributed.

Figure 4: Quantitative evaluation of the distribution differences on benchmark: CIFAR-10 Interpolation, CIFAR-100, SVHN and CelebA with KS and Cohen's $d$ statistics.

### 4.4 OUT-OF-DISTRIBUTION-DETECTION

Energy-based models (EBMs), by design, incorporate an intrinsic informative energy prior that makes them particularly well-designed for out-of-distribution (OOD) detection. Leveraging this property, our pretrained EBM can be directly employed to OOD tasks without further fine-tuning. Specifically, given the final energy model in the pipeline, we evaluate both in-distribution and out-of-distribution samples and compute the AUROC scores, with results reported in Table 3. Remarkably, the model demonstrates strong performance across standard benchmarks, including CIFAR-10 interpolation, CelebA, and SVHN. To further highlight the differences between the energy distributions, we compute and visualize Kolmogorov–Smirnov (KS) statistics and Cohen's $d$ scores, as shown in Figure 4. The trend aligns with AUROC results as well. Additional demonstrations with explicit energy notations are provided in the appendix to facilitate more detailed comparisons.

## 5 CONCLUSION

We introduced VDRME, a scalable framework for training multiscale energy-based models using variational diffusion recovery and entropy-regularized generators. By amortizing Langevin MCMC with conditional generators, VDRME enables efficient one-step sampling per scale, significantly reducing inference cost. Unlike diffusion GAN models, our approach maintains a fully energy-based formulation, while benefiting from the inductive bias of diffusion models. The use of entropy regularization enhances sample diversity and stability, addressing common challenges such as mode collapse. Empirical results demonstrate that VDRME achieves fast image generation, while outperforming EBMs with the same computational complexity and being comparable to state-of-the art generative models. It also demonstrates its utility in downstream tasks including out-of-distribution detection and density estimation. These findings position VDRME as a principled and efficient alternative to traditional EBMs and diffusion-based generative models. Our work

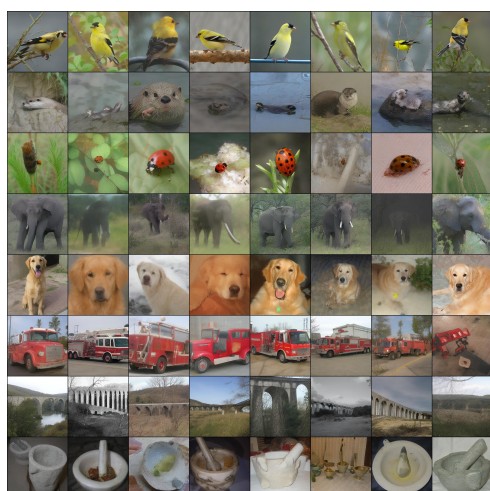

Figure 5: Class-conditional samples generated by VDRME on ImageNet-256p with 4 sampling steps. Each row corresponds to samples from the same class. The model achieves an FID score of 4.62.

aims to stimulate further research on developing EBMs as generative models. However, the prevalence of powerful generative models may give rise to negative social consequences, such as deepfakes, misinformation, privacy breaches, and erosion of public trust, highlighting the need for effective preventive measure.

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

# A APPENDIX

## A.1 LITERATURE REVIEW

## A.2 EXTENDED DERIVATION

### A.2.1 DERIVATION OF EQUATION 4

$$
\theta = \arg\min_\theta \; \Big( \underbrace{\int p_{\text{data}}(x) \log p_{\text{data}}(x) dx + \log Z_\theta)}_{\mathbb{E}_{x \sim p_{\text{data}}}[E_\theta(x)]}
$$

$$
= \arg\min_\theta \; \Big( \mathbb{E}_{x \sim p_{\text{data}}}[E_\theta(x)] - \underbrace{\int p_\theta(x) \log p_\theta(x) dx}_{\mathbb{E}_{x \sim p_\theta}[E_\theta(x)]} \Big)
$$

(20)

### A.2.2 DERIVATION OF EQUATION 9

Given a variational diffusion $q_\phi(x_t|x_{i+1}, t)$, the equality holds when the variational distribution equals the conditional energy distribution $p_\theta(x_t|x_{t+1}, t)$. We can estimate the normalization term by:

$$
\log Z_\theta(t_i) = \log \int q_\phi(x_i|x_{i+1}, t) \frac{\exp(-\mathcal{E}_\theta(x_i|x_{i+1}, t))}{q_\phi(x_i|x_{i+1}, t)} dx_{t_i}
$$

$$
\geq \int q_\phi(x_i|x_{i+1}, t_i) \log \frac{\exp(-\mathcal{E}_\theta(x_i|x_{i+1}, t_i))}{q_\phi(x_t|x_{i+1}, t_i)} dx
$$

(21)

$$
\geq -\mathbb{E}_{x_{t_i} \sim q_\phi(x_{i+1}, t)}\Big[\mathcal{E}_\theta(x_i|x_{i+1}, t_i)\Big] - \mathbb{E}_{x_{t_i} \sim q_\phi(x_{i+1}, t_i)}\Big[\log q_\phi(x_i|x_{i+1})\Big]
$$

The last expression is a lower bound for $\log Z_\theta(t_i)$ for all $q_\phi$, which implies that

$$
\log Z_\theta(t_i) \geq \max_\phi \Big( -\mathbb{E}_{x_{t_i} \sim q_\phi(x_{i+1}, t_i)}[\mathcal{E}_\theta(x_i|x_{i+1}, t_i)] + \underbrace{-\mathbb{E}_{q_\phi(x_{i+1}, t)}[\log q_\phi(t_i)]}_{H[q_\phi(x_{i+1}, t_i)]} \Big)
$$

### A.2.3 DERIVATION OF EQUATION 15

To reverse the diffusion path, we parametrize $G_\phi$ with current noisy sample $x_{t+1}$, the auxiliary variable $z_i$, and the time-scale $t$,

$$
x_{t_i} = \sqrt{\tilde{\alpha}_{t_i}} \underbrace{\left( \frac{x_{i+1} - \sqrt{1 - \tilde{\alpha}_{t_{i+1}}} \; \epsilon_\phi(x_{i+1}, z_i, t_i)}{\sqrt{\tilde{\alpha}_{t_{i+1}}}} \right)}_{\hat{x}_0} + \sqrt{1 - \tilde{\alpha}_{t_i}} \; \epsilon_\phi(x_{i+1}, z_i, t_i) \quad (22)
$$

$$
= \underbrace{\frac{x_{i+1}}{\sqrt{\alpha_{t_i}}} + \left( \sqrt{1 - \tilde{\alpha}_{t_i}} - \frac{\sqrt{1 - \tilde{\alpha}_{t_{i+1}}}}{\sqrt{\alpha_{t_i}}} \right) \; \epsilon_\phi(x_{i+1}, z_i, t_i)}_{G_\theta(x_{t_{i+1}, z_i, t_i})} \quad (23)
$$

$$
= \underbrace{\frac{1}{\sqrt{1 - \beta_{t_i}}} \Big( x_{t+1} - \sqrt{\tilde{\beta}_{t_{i+1}}} \epsilon_\phi(x_{i+1}, z_i, t_i) \Big) + \sqrt{\tilde{\beta}_{t_i}} \epsilon_\phi(x_{i+1}, z_i, t_i)}_{G_\phi(x_{t_{i+1}, z_i, t_i})} \quad (24)
$$

Unlike DDPM, which relies on stochastic Gaussian sampling at every step, or DDIM, which enforces a deterministic trajectory through reparameterization, our formulation integrates the predictor $\epsilon_\phi$ directly into a parametric generator function and injects randomness into the generator.

## A.3 TRAINING DETAILS

### A.3.1 NETWORK ARCHITECTURES

To accelerate training, we initialize our framework with a widely adopted pretrained backbone with zero initialization of the last layer of the energy model. As noted in prior work (Grathwohl et al., 2019a), the generator can be viewed as a classifier in its functional form, and the feature representations often share similarities. Motivated by this, we adopt the same core architecture for both the generator and the energy model. Following the design used in entropy-based estimators (Kumar et al., 2019), we extend the energy model by adding an auxiliary classifier on top of the backbone. Consequently, the generator, the energy model, and the entropy estimator all share the same main architecture in our implementation.

To incorporate noise input, we introduce an additional ResNet block equipped with both time and class embeddings (if applicable), consistent with the backbone design. This block is first pretrained to disregard noise perturbations before being connected to the main network, which substantially stabilizes training. For the energy model and entropy estimator, we append lightweight linear layers that project the backbone output to a scalar, representing the energy value. On top of this, we further add a classifier to distinguish between true and fake pairs with respect to the noise input variable $z_i$.

For CIFAR-10, we adopt a pretrained U-Net backbone originally trained with DDPM (Ho et al., 2020a). For ImageNet-256, we use the DiT-XL/2 architecture as the backbone. In the case of low-dimensional data, such as the checkerboard experiment, we follow the architecture described in (Zhu et al., 2023), with the modification that noise is pre-conditioned into the generator.

### A.3.2 HYPERPARAMETER RECOMMENDATIONS

Table 4: Hyperparameters used in our experiments.

| Type | Hyperparameter | Value |
|---|---|---|
| Optimization | $(\beta_1, \beta_2)$ (energy) | (0, 0.999) |
| | $(\beta_1, \beta_2)$ (generator) | (0.3, 0.999) |
| | $(\beta_1, \beta_2)$ (entropy) | (0.3, 0.999) |
| | learning rate (energy) | $4 \cdot 10^{-5}$ |
| | learning rate (generator) | $2 \cdot 10^{-5}$ |
| | learning rate (entropy) | $1 \cdot 10^{-5}$ |
| | eps | $1 \cdot 10^{-12}$ |
| Weights | $\lambda_e$ (entropy regularization weight) | 1 |
| | $\lambda_g$ (gradient norm penalty) | 0.01 |
| | $w_e$ (energy weight) | 1 |
| | $w_r$ (diffusion recovery weight) | 1 |

After extensive experimentation, we recommend the above set of hyperparameters as the most suitable configuration for our method. A lower value of $\beta_1$ combined with a relatively higher learning rate allows the energy model to react more rapidly to the updates of the generator, thereby stabilizing the training dynamics. The weighting coefficients are aligned with the formulation presented in the main paper and have been empirically validated to yield the best overall performance.

Table 5: Comparison of FID scores with public models on CIFAR-10.

| Models | Steps | FID ↓ |
|---|---|---|
| **Explicit EBM based method** | | |
| NT-EBM (Nijkamp et al., 2020) | - | 78.12 |
| LP-EBM (Pang et al., 2020) | 40 | 70.15 |
| Adaptive CE (Xiao & Han, 2022) | 40 | 65.01 |
| JEM (Grathwohl et al., 2019b) | - | 38.40 |
| EBM-IG (Du & Mordatch, 2019) | - | 38.20 |
| EBM-FCE (Gao et al., 2020a) | - | 37.30 |
| CoopVAEBM (Xie et al., 2021) | 15 | 36.20 |
| Divergence Triangle (Han et al., 2020) | - | 30.10 |
| VARA (Grathwohl et al., 2020) | - | 27.50 |
| EBM-CD (Du et al., 2020) | 40 | 25.10 |
| GEBM (Arbel et al., 2020) | - | 19.31 |
| HAT-EBM (Hill et al., 2022) | 50 | 19.30 |
| CF-EBM (Zhao et al., 2020b) | 60 | 16.71 |
| CoopFlow (Xie et al., 2022) | 30 | 15.80 |
| CLEL-base (Lee et al., 2023a) | 600 | 15.27 |
| VAEBM (Xiao et al., 2020) | 16 | 12.16 |
| CDRL (Guo et al., 2023b) | 18 | 9.67 |
| DRL (Gao et al., 2020c) | 180 | 9.58 |
| CLEL-large (Lee et al., 2023a) | 1200 | 8.61 |
| CDRL (Guo et al., 2023b) | 90 | 4.31 |
| Distilled-EnergyDiffusion (Thornton et al., 2025) | 35 | 3.01 |
| **VDRME (Ours)** | **4** | **7.83** |
| **VDRME (Ours)** | **6** | **7.62** |
| **VDRME (Ours)** | **8** | **6.90** |

| Models | FID ↓ |
|---|---|
| **Other likelihood based method** | |
| VAE (Kingma & Welling, 2013) | 78.41 |
| PixelCNN (Salimans et al., 2017) | 65.93 |
| PixelIQN (Ostrovski et al., 2018) | 49.46 |
| Residual Flow (Chen et al., 2019) | 47.37 |
| Glow (Kingma & Dhariwal, 2018) | 45.99 |
| DC-VAE (Parmar et al., 2021) | 17.90 |
| **GAN based method** | |
| WGAN-GP (Gulrajani et al., 2017) | 36.40 |
| SN-GAN (Miyato et al., 2018) | 21.70 |
| BigGAN (Brock et al., 2018) | 14.80 |
| StyleGAN2-DiffAugment (Zhao et al., 2020a) | 5.79 |
| Diffusion-GAN (Xiao et al., 2021) | 3.75 |
| StyleGAN2-ADA (Karras et al., 2020) | 2.92 |
| **Score based and Diffusion method** | |
| NCSN (Song & Ermon, 2019) | 25.32 |
| NCSN-v2 (Song & Ermon, 2020) | 10.87 |
| DDPM Distillation (Luhman & Luhman, 2021) | 9.36 |
| DDIM (T=20) (Song et al., 2020a) | 6.84 |
| DDPM++(VP, NLL) (Kim et al., 2021) | 3.45 |
| DDPM (Ho et al., 2020b) | 3.17 |
| DDPM++(VP, FID) (Kim et al., 2021) | 2.47 |
| EDM (Karras et al., 2022) | 2.21 |
| NCSN++ (Song et al., 2021) | 2.20 |

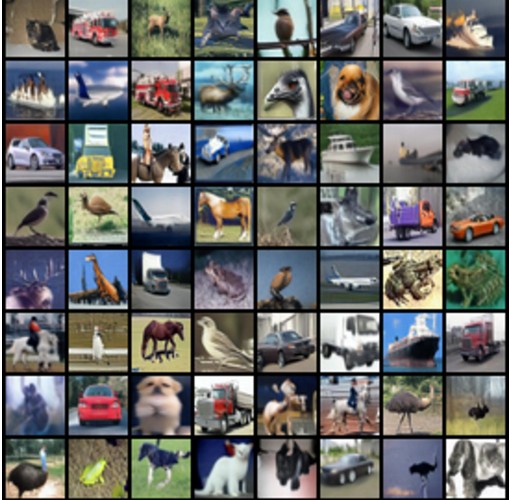 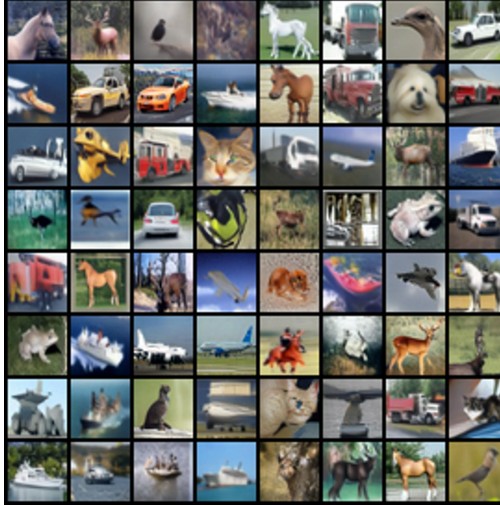

Figure 6: Unconditional generation on CIFAR-10 with 4 steps sampling, FID = 7.83.

## A.4 MORE EXPERIMENTS DETAILS

Despite the remaining performance gap between EBMs and the most recent advances in few-step generation, the energy model provides a versatile framework for building multi-task model zoos. Beyond its primary role in guided and unguided image generation, EBMs can be naturally extended to a variety of tasks, including inverse problems, density estimation, and out-of-distribution detection. For illustration, we demonstrate applications of VDRME to both conditional and unconditional image generation, density estimation, and OOD detection.

### A.4.1 IMAGE GENERATION

For the case of unconditional generation, we compare more results shown in Table 5. VDRME has fewer sampling steps. Meanwhile, we take a 4-step conditional generation on ImageNet-256p.

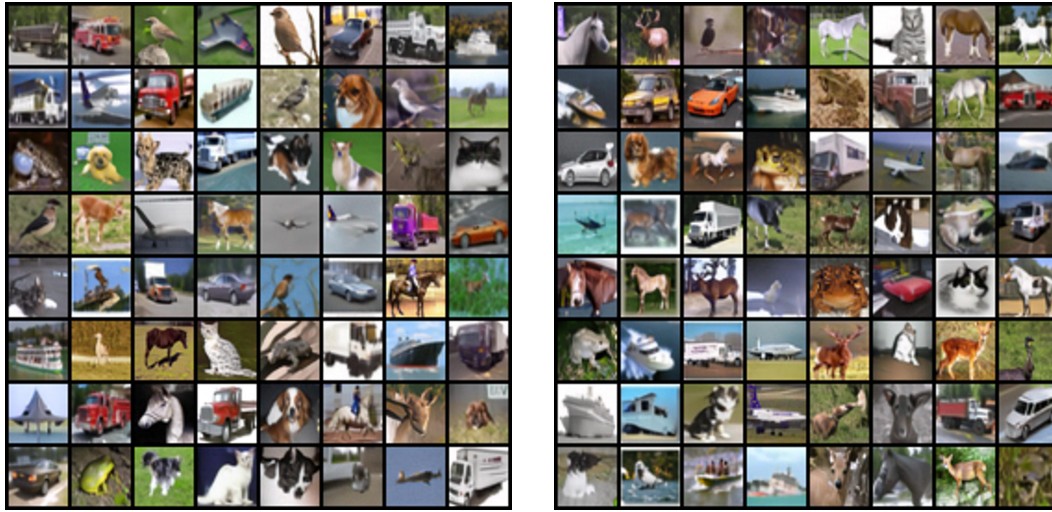

Figure 7: Unconditional generation on CIFAR-10 with 6 steps sampling, FID = 7.62.

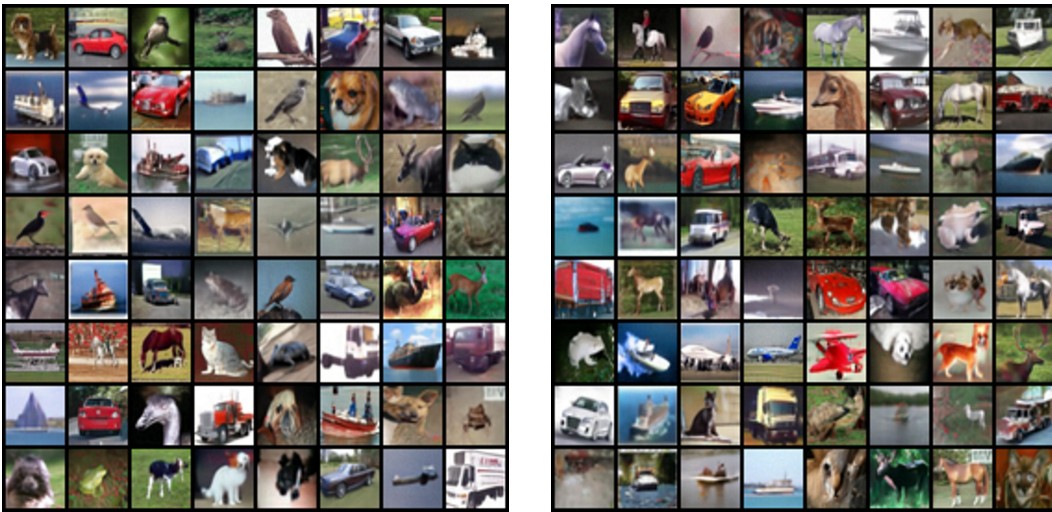

Figure 8: Unconditional generation on CIFAR-10 with 8 steps sampling, FID = 6.90.

### A.4.2 OUT-OF-DISTRIBUTION DETECTION

We analyze the energy of out-of-distribution samples alongside the corresponding generated images. The energy scores are evaluated using only the final-stage energy model.

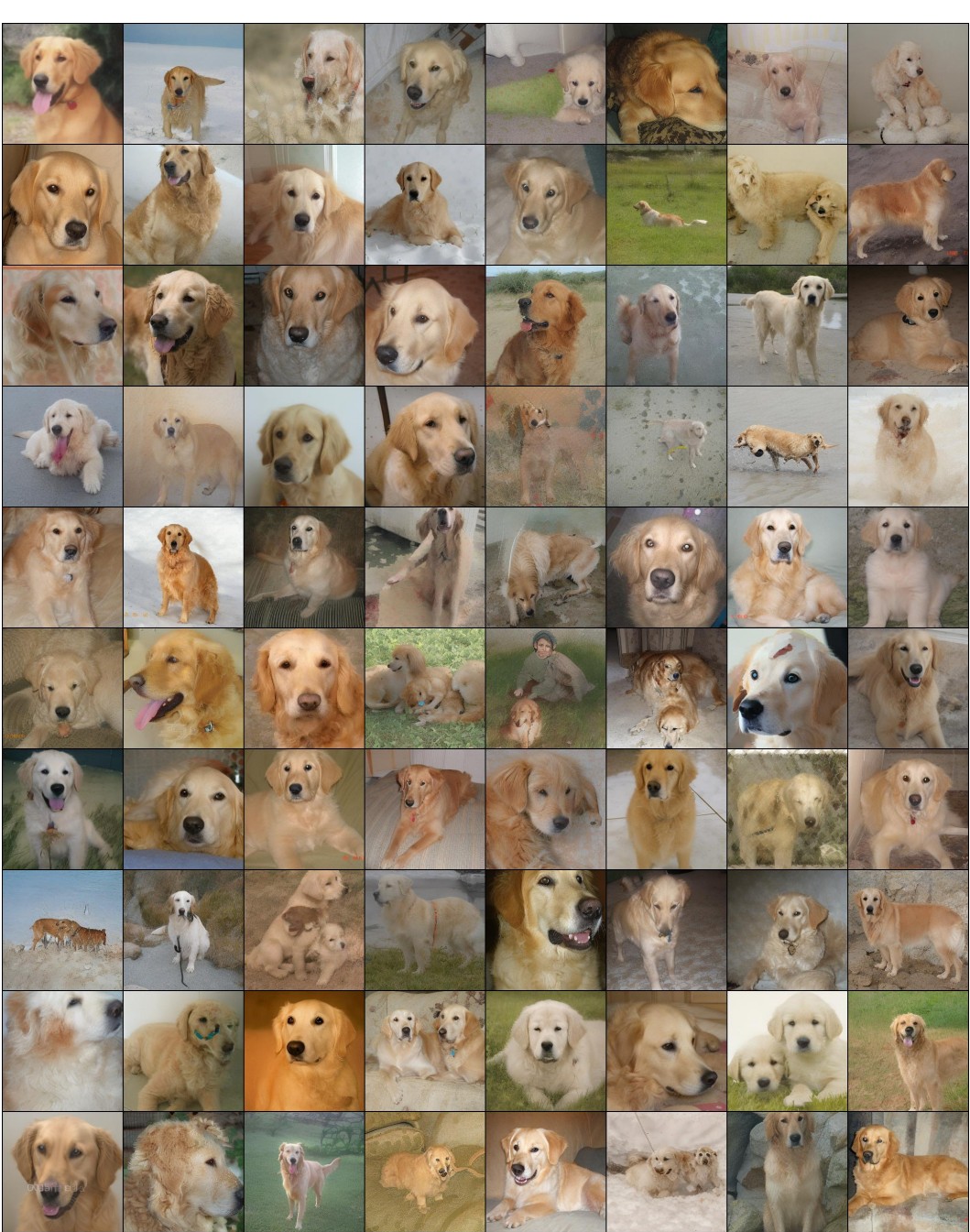

Figure 9: Conditional generation (single class) on ImageNet-256p with 4 steps sampling, FID = 4.62.

1026
1027
1028
1029
1030
1031
1032
1033
1034
1035
1036
1037
1038
1039
1040
1041
1042
1043
1044
1045
1046
1047
1048
1049
1050
1051
1052
1053
1054
1055
1056
1057
1058
1059
1060
1061
1062
1063
1064
1065
1066
1067
1068
1069
1070
1071
1072
1073
1074
1075
1076
1077
1078
1079

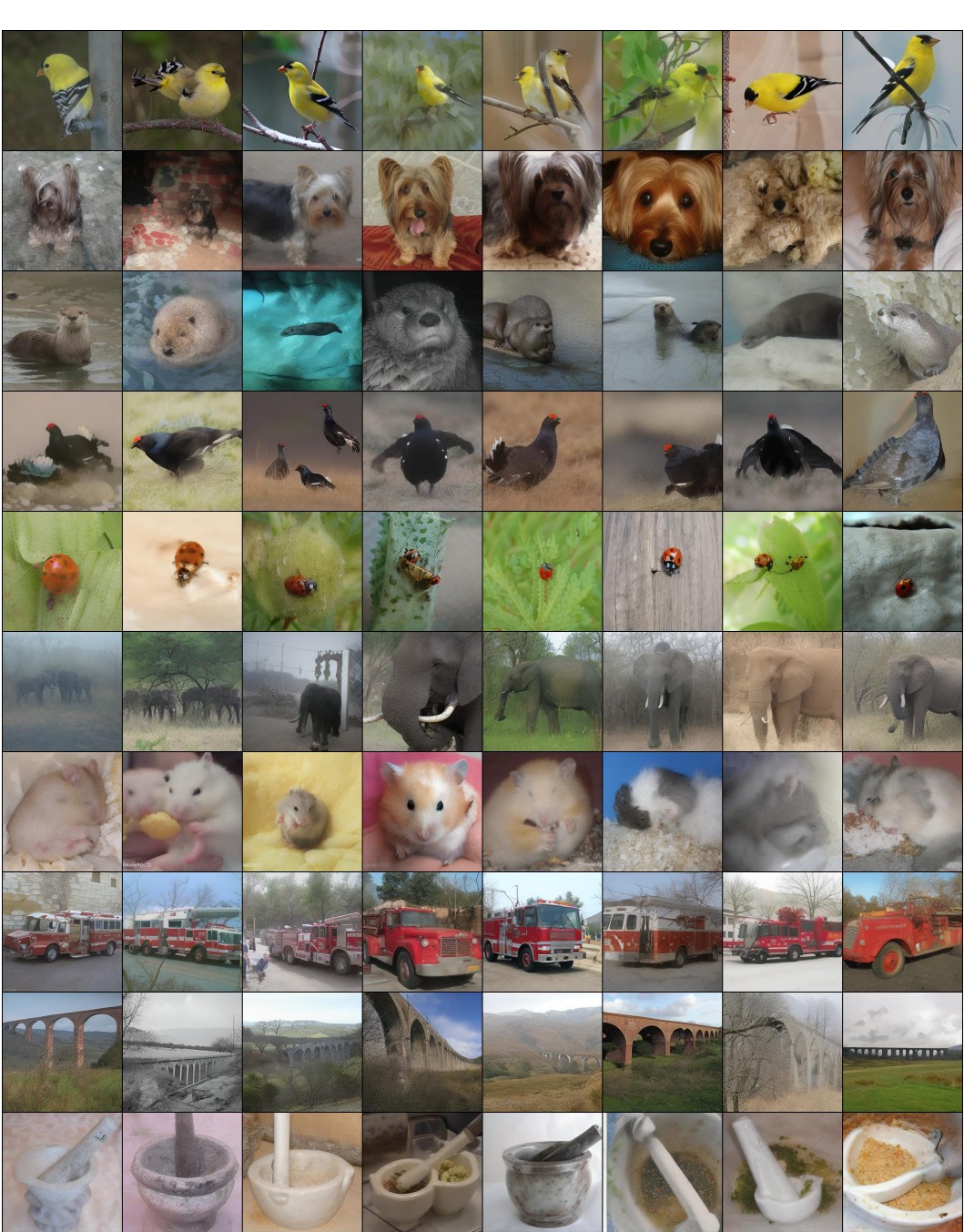

Figure 10: Conditional generation on ImageNet-256p with 4 steps sampling, FID = 4.62.Each row corresponds to samples from a specific class

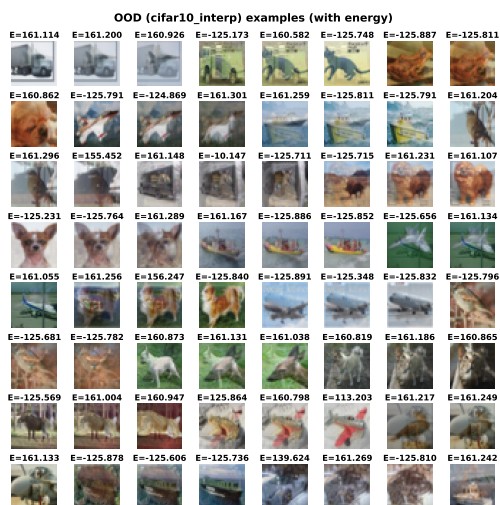

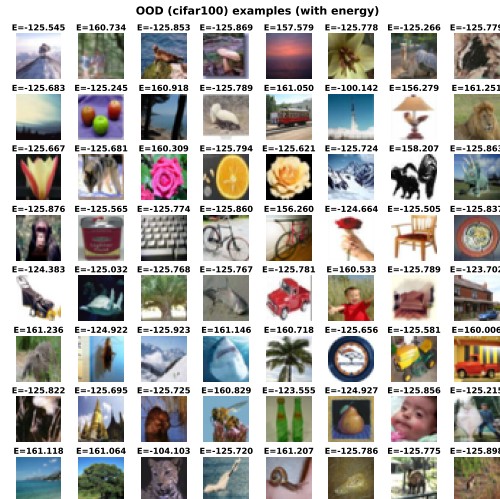

Figure 11: Energy of the CIFAR-10 interpolation, AUROC = 0.72.

Figure 12: Energy of the CIFAR-100, AUROC = 0.54.

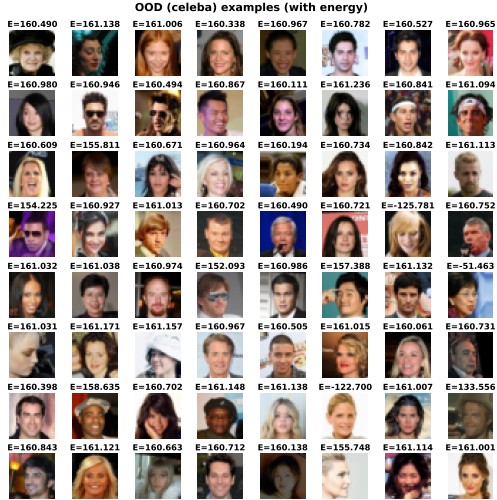

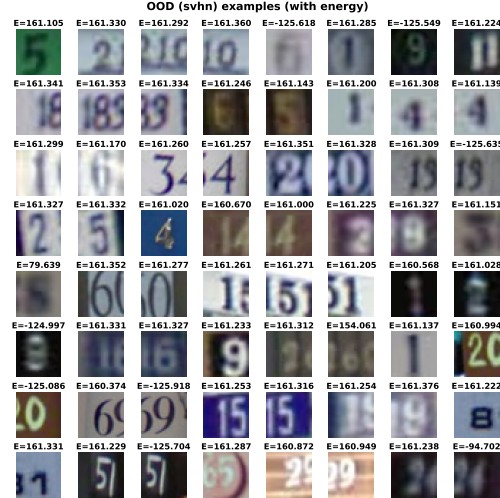

Figure 13: Energy of the CelebA, AUROC = 0.84.

Figure 14: Energy of the SVHN, AUROC = 0.92.

