# OpenReview forum: "Variational Diffusion Recovery with Multiscale Energy Models for Few-Step Generation"
_ICLR.cc/2026/Conference — ICLR 2026 Conference Withdrawn Submission_

### Official Review · Reviewer_e1ek · 2025-10-31

**Soundness:** 2
**Presentation:** 1
**Contribution:** 2
**Rating:** 4
**Confidence:** 4

**Summary:**

This paper introduces VDRME, a novel framework for training multiscale energy-based models (EBMs) that achieves fast, few-step generation. The core contribution is to replace the slow and often unstable MCMC sampling required by traditional EBM training methods (like contrastive divergence or diffusion recovery likelihood) with amortized sampling via a conditional generator at each noise scale. This is achieved by reformulating the maximum likelihood objective as a bi-level variational optimization problem. The authors introduce an entropy regularization term on the generator to ensure sample diversity and prevent mode collapse. The result is a principled, fully energy-based model that generates high-quality images in very few steps (4-8), achieving state-of-the-art performance among EBMs while being competitive in speed with modern diffusion and flow-based models.

**Strengths:**

1.  VDRME provides an elegant solution to the long-standing problem of slow sampling in EBMs. By amortizing MCMC with a conditional generator within a variational framework, it bridges the gap between the theoretical appeal of EBMs and the practical necessity of efficient training /generation.

2.  The paper successfully demonstrates that the learned energy function is not just a byproduct of training but is highly effective for downstream tasks. The strong performance on out-of-distribution (OOD) detection without any fine-tuning is a key advantage of the EBM formulation over other generative models.

**Weaknesses:**

1.  The method relies on initializing the backbone with pre-trained DDPM weights to stabilize training. While this is a practical choice, an ablation study showing the performance difference when training from scratch would provide a clearer picture of the framework's inherent stability.

2.  The approach has conceptual similarities to the "Diffusion-GAN". While the paper notes key differences (e.g., WGAN-like loss, no classifier), a more detailed discussion and comparison in the related work section would help to better situate the contribution.

3. The tables in this article are so ugly.

**Questions:**

N/A

---

### Official Review · Reviewer_hKuh · 2025-11-01

**Soundness:** 2
**Presentation:** 2
**Contribution:** 2
**Rating:** 4
**Confidence:** 3

**Summary:**

This paper proposes VDRME, a framework that combines energy-based models (EBMs) with diffusion recovery likelihood to enable one-step sampling per scale. The key innovation is replacing the MCMC sampling at each noise scale with learned conditional generators. The generators are regularized with entropy-based terms to prevent mode collapse. The author conducts experiments on CIFAR-10 dataset and ImageNet-256p.

**Strengths:**

Using a variational lower bound on the maximum likelihood objective to enable efficient one-step sampling per scale for EBM is novel.

**Weaknesses:**

1. The training cost of the proposed method is expensive, while the benefit is relatively small. To train the proposed EBM, the author requires a pre-trained U-Net from DDPM. And then the algorithm requires training three networks: EBM, diffusion model, and mutual information estimator. However, the performance of the trained EBM achieves worse performance than the pre-trained U-Net from DDPM.

2. Appendix A.1 "LITERATURE REVIEW" is missing.

3. Appendix A.2.1 "DERIVATION OF EQUATION 4" is wrong. The first line of Eq (20) is wrong: the first term should be $\int p_{data}(x) E_\theta(x) dx$. The second line of Eq (20) might need more proof: whether $\log Z_\theta = E_{x\sim p_{\theta}}[E_{\theta}(x)]$. Therefore, I cannot understand the beginning of section 3.1, i.e., how to extend Eq (4) to Eq (7).

**Questions:**

See Weaknesses 1, 3.

---

### Official Review · Reviewer_WScr · 2025-11-01

**Soundness:** 3
**Presentation:** 2
**Contribution:** 2
**Rating:** 4
**Confidence:** 4

**Summary:**

The paper proposes VDRME that accelerates training and inference of multiscale energy-based models (EBMs) by replacing iterative Langevin MCMC at each diffusion recovery step with a conditional generator, amortizing sampling. Training is a minimax game between the EBM and generator with an entropy regularizer to promote diversity. Experiments on CIFAR-10 and ImageNet show fast sampling (4–8 steps) and improvements over prior EBMs in FID and efficiency.

**Strengths:**

1. The paper reinterprets the multiscale maximum likelihood objective as a variational optimization with amortized samplers to effectively address the EBM efficiency bottleneck.

2. The learned energy function is useful for OOD detection, and shows clear advantage over current one-/few-step generation methods such as GANs and score distillation.

**Weaknesses:**

1. The practical motivation and benefits of VDRME are vague. First, although VDRME obtains stronger results compared to EBMs, it is still inferior to state-of-the-art few-step methods like consistency models, mean flows, and score distillation. Moreover, the stability of the proposed framework seems to heavily depend on initializing from a powerful pre-trained DDPM. This is a significant dependency that limits the generalizability of the proposed framework.

2. Experimental settings and explanation are insufficient. For example, the experimental setting in Section 3.3 is unclear. The explanation of AUROC calculation is insufficient. It is important to clearly describe and explained the reasonability (e.g., commonly adopted settings following previous works) of experiment setup for a convincing evaluation on the proposed framework.

3. Amortizing the diffusion recovery process at different noise levels has already been considered in recent works like [R1]-[R3]. Discussion about difference and empirical comparison are necessary to clarify the contributions of this paper.

4. The calculation of log-likelihood is not clarified and the quantitive comparison of log-likelihood in terms of bits per dim with likelihood-based models like VDM [R4] and pixel CNN [R5] is missing.

[R1] Valentin De Bortoli, Alexandre Galashov, J Swaroop Guntupalli, Guangyao Zhou, Kevin Patrick Murphy, Arthur Gretton, Arnaud Doucet. Distributional Diffusion Models with Scoring Rules. ICML 2025.

[R2] Xinyu Peng, Ziyang Zheng, Yaoming Wang, Han Li, Nuowen Kan, Wenrui Dai, Chenglin Li, Junni Zou, Hongkai Xiong. Noise Conditional Variational Score Distillation. ICML 2025.

[R3] Zhisheng Xiao, Karsten Kreis, and Arash Vahdat. Tackling the generative learning trilemma with denoising diffusion GANs. ICLR 2022.

[R4] Diederik P. Kingma, Tim Salimans, Ben Poole, Jonathan Ho. Variational diffusion models. NeurIPS 2021.

[R5] Tim Salimans, Andrej Karpathy, Xi Chen, and Diederik P Kingma. PixelCNN++: Improving the pixelcnn with discretized logistic mixture likelihood and other modifications. ICLR 2017.

**Questions:**

Please refer to the section of Weaknesses.

---

### Official Review · Reviewer_aQUq · 2025-11-01

**Soundness:** 2
**Presentation:** 2
**Contribution:** 2
**Rating:** 4
**Confidence:** 4

**Summary:**

This work proposes VDRME, a hybrid framework that combines energy-based models (EBMs) and diffusion models to address the sampling inefficiency of traditional EBMs. The core idea of amortizing MCMC via variational diffusion recovery is intriguing, with strong empirical results on image generation and downstream tasks (out-of-distribution detection, density estimation).

**Strengths:**

1. The paper identifies sampling inefficiency (Langevin MCMC dependency) as a key bottleneck for EBMs, and its focus on few-step generation aligns with practical needs in generative modeling.

2. The results on CIFAR-10 (8-step FID=6.90) and ImageNet 256 (4-step FID=4.62) show that the proposed framework can generate competitive-quality samples and confirm the feasibility of amortizing MCMC via generators.

3. The proposed method retains the strengths of EBMs in downstream tasks (e.g., out-of-distribution detection) by maintaining a fully energy-based formulation.

**Weaknesses:**

1. The core ideas of variational EBM optimization, generator-based MCMC amortization, and entropy regularization have been already considered in existing works [R1]-[R3]. This paper is an adjustment and combination of these existing ideas for the goal of adapting to multi-scale generation.

2.  The design of VDRME is incremental in technical contributions compared to existing methods [R1]-[R3]. Technical difficulties and innovations are not clarified.

i) Grathwohl et al. [R1] introduced the minimax game between the energy function and the generator as an alternative to MCMC sampling in traditional EMBs, and Gao et al. [R2] achieved multi-scale diffusion to split EBM training and reduced sampling difficulty with conditional EBM but resort to MCMC. VDRME combines the minimax game and multi-scale diffusion, and uses a single-step generator for each noise scale.

ii)  Kumar et al. [R3] implemented entropy regularization to prevent the generator pattern from crashing, while VDRME changed single-scale mutual information to calculate mutual information for each scale separately.

3. VDRME yields degraded performance compared to state-of-the-art few-step generative models (e.g., Consistency Models on CIFAR-10) and fails to demonstrate unique advantages over diffusion models for downstream tasks (e.g., AUROC of 0.92 on SVHN vs 0.98 on SVHN by the prior EBM CLEL in OOD detection).

[R1] Will Grathwohl, Kuan-Chieh Wang, Joern-Henrik Jacobsen, David Duvenaud, Mohammad Norouzi, Kevin Swersky, "Your classifier is secretly an energy based model and you should treat it like one," in ICLR 2020.

[R2] Rithesh Kumar, Sherjil Ozair, Anirudh Goyal, Aaron Courville, Yoshua Bengio, "Maximum Entropy Generators for Energy-Based Models," 	arXiv:1901.08508, 2019.

[R3] Yaxuan Zhu, Jianwen Xie, Ying Nian Wu, Ruiqi Gao, "Learning energy-based models by diffusion recovery likelihood," in ICLR 2021.

**Questions:**

Please refer to Weaknesses.

---

### Note · Authors · 2025-12-02

I have read and agree with the venue's withdrawal policy on behalf of myself and my co-authors.